# Long-Term IGF-1 Maintenance in the Upper-Normal Range Has Beneficial Effect on Low-Grade Inflammation Marker in Adults with Growth Hormone Deficiency

**DOI:** 10.3390/ijms26052010

**Published:** 2025-02-25

**Authors:** Ana Klinc, Andrej Janež, Mojca Jensterle

**Affiliations:** 1Faculty of Medicine, University of Ljubljana, 1000 Ljubljana, Slovenia; anaa.klinc@gmail.com (A.K.); andrej.janez@kclj.si (A.J.); 2Department of Endocrinology, Diabetes and Metabolic Diseases, University Medical Center Ljubljana, 1000 Ljubljana, Slovenia

**Keywords:** adult growth hormone deficiency, growth hormone replacement therapy, IGF-1 range, high-sensitivity C-reactive protein, body composition

## Abstract

The distinctive effects of maintaining the upper- (0–2) versus lower-normal (−2–0) range of IGF-1 SDS in adult growth hormone deficiency (AGHD) remain understudied. We conducted a cross-sectional study on 31 patients with AGHD receiving growth hormone replacement therapy (GHRT) with daily GH for >5 years, with a 2-year mean IGF-1 SDS ranging between −2 and +2. Patients were categorized into the upper- or lower-normal range IGF-1 SDS groups according to their 2-year mean. Associations of clinical characteristics, anthropometric parameters, laboratory tests, and vascular markers of subclinical atherosclerosis with the 2-year IGF-1 SDS range and 5-year mean IGF-1 SDS were explored. Long-term maintenance of upper-normal IGF-1 SDSs was more common in men and in patients with a longer duration of GHRT. Patients with tumor-related AGHD had a lower 5-year mean IGF-1 SDS. Long-term maintenance of IGF-1 SDS in the upper-normal range was associated with lower high-sensitivity C-reactive protein (hs-CRP) levels (median (25–75% range): 0.8 (0.6–1.1) vs. 1.8 (0.8–4.6); *p* = 0.005). Moreover, a negative correlation was identified between a hs-CRP and the 5-year mean IGF-1 SDS. The association between the upper-normal IGF-1 SDS range and lower body fat percentage lost significance after adjusting for sex, due to the higher proportion of male patients in the upper-normal IGF-1 SDS group. In conclusion, long-term maintenance of upper-normal IGF-1 SDSs was associated with male sex and reduced low-grade inflammation. Randomized controlled studies are needed to evaluate the long-term and sex-specific effects of targeting the upper- vs. lower-normal IGF-1 range in AGHD.

## 1. Introduction

Adult growth hormone deficiency (AGHD) is characterized by visceral obesity, reduced lean body mass, proatherogenic lipid profile, and insulin resistance [1]. Moreover, a chronic pro-inflammatory state, increased oxidative stress, and an impaired adipokine profile are linked to AGHD [1,2]. These cardiovascular risk factors contribute to the accelerated progression of atherosclerosis in patients with AGHD, evidenced by a higher prevalence of subclinical atherosclerosis markers [3], including endothelial dysfunction [4] and increased carotid intima-media thickness (IMT) [3]. Of relevance is the fact that increased cardiovascular morbidity and mortality in patients with hypopituitarism is mainly related to growth hormone (GH) deficiency [5,6].

Most cardiovascular risk factors associated with AGHD are potentially reversible using growth hormone replacement therapy (GHRT) [1,3,7]. Nevertheless, controversy persists regarding the appropriate use of GHRT, including the burden of daily injections, lack of awareness of its benefits, the vague and subtle array of symptoms in GHD, and concerns over potential adverse effects [8,9]. Compliance represents a major issue in GHRT, with high discontinuation rates and a progressive decline in adherence over time [8]. In addition, it is not clear which patient characteristics may further influence the effectiveness of GHRT. In view of all these challenges, the efficacy of GHRT in reversing cardiovascular risk and atherosclerosis progression remains limited.

Over the past decades, dosing regimens have evolved from supraphysiological weight-based doses to fixed lower doses and finally to individually titrated doses according to IGF-1 levels, clinical response, and side effects [10]. To date, age- and sex-adjusted IGF-1 standard deviation scores (SDSs) remain the most widely used biomarker for monitoring GHRT in clinical practice, regardless of their weak correlation with clinical endpoints in GHRT [9,11]. The generally accepted recommendation for titrating GHRT is to reach an IGF-1 SDS within the normal reference range of –2 to +2 [11]. Little is known about potential differences in the efficacy and safety of GHRT depending on whether IGF-1 SDS is maintained in the lower or upper half of the normal range. The target is commonly set in the upper-normal range [10], as emerging evidence suggests high-normal IGF-1 SDSs (1 to 2 SD) may be associated with a more favorable body composition and wellbeing [12,13]. Currently, data on the optimal target IGF-1 SDS range are scarce. In addition, the effects of GHRT and the extent to which they are dose-dependent and thus IGF-1-related represent an understudied entity.

We aimed to investigate the effects of maintaining the upper- (0–2) vs. lower-normal (−2–0) IGF-1 SDS range in patients with AGHD undergoing treatment with GHRT for at least 5 years and having a 2-year mean IGF-1 SDS between −2 and +2. Associations of long-term maintenance of IGF-1 SDS in the upper- or lower-normal range with patient characteristics, cardiovascular risk factors, and vascular parameters were explored. In addition, we analyzed which effects of GHRT potentially exhibit a direct correlation with the 5-year mean IGF-1 SDS. We hypothesized that patients with upper-normal IGF-1 SDSs would exhibit a more favorable cardiovascular profile compared to those with IGF-1 SDSs within the lower-normal range.

## 2. Results

### 2.1. Patient Characteristics

Continuous variable distributions for clinical and anthropometric characteristics, as well as vascular parameters, characteristics of the underlying pituitary disease, and its treatment for the entire group, are presented in Table 1. Our cohort included 14 women and 17 men, with a mean age of 46.3 (range: 22–68) years. Etiologically, 8 patients had non-tumor-related GHD, while 23 patients had tumor-related etiology. None of the latter had residual or recurrent tumors, and four had a history of craniopharyngioma. None of the patients had isolated GHD; other pituitary hormone deficiencies were optimally supplemented according to the patients’ needs. Among the 14 female patients, all 9 premenopausal women were receiving oral estrogen replacement, while none of the postmenopausal women were on hormone replacement therapy. The mean duration of GHRT in our patients was 20.9 years, with a range of 5 to 47 years. In terms of lifestyle and comorbidities, 6 patients were active or former smokers, 5 patients had arterial hypertension, and 8 patients were on antihyperlipidemic agents. None of the patients had diabetes mellitus. Over the period of two years, 18 patients maintained upper-normal IGF-1 SDSs, while 13 patients had lower-normal levels.

### 2.2. Long-Term IGF-1 SDS Range in Relation to Patient Characteristics

The variables tested to determine their association with the long-term maintenance of IGF-1 SDS in the upper- vs. lower-normal range are presented in Table 2. Following correction for multiple testing, sex was found to be significantly associated with the 2-year IGF-1 SDS range, as women were less likely to be within the upper-normal range. The effect of sex was also evident over the 5-year period, as the median female 5-year mean IGF-1 SDS (median −0.27 SD, 25–75% range −1.46–1.19 SD) fell into the lower-normal range while males (median 1.04 SD, 25–75% range 0.38–1.33 SD; *p* = 0.039) reached upper-normal levels. Patients with upper-normal IGF-1 SDSs tended to have a longer duration of GHRT administration. Furthermore, a positive correlation between the 5-year mean IGF-1 SDS and the duration of GHRT was nominally significant (r = 0.421, *p* = 0.036). Patients with tumor-related GHD had a lower 5-year mean IGF-1 SDS compared to those with non-tumor-related etiology (median (25–75% range): 0.08 SD (−1.35–1.15) vs. 1.25 SD (0.97–1.66); *p* = 0.006), although the median values for both groups were within the upper-normal range. The proportion of tumor-related etiology was lower in the group with upper-normal IGF-1 SDSs compared to those with lower-normal levels; however, the difference was not significant. Patients receiving oral estrogen were relatively evenly distributed between the two IGF-1 SDS range groups, with 4 in the upper-normal and 5 in the lower-normal range. On the other hand, women not receiving estrogen replacement were less evenly distributed, with 4 out of 5 in the lower-normal IGF-1 SDS range.

### 2.3. Long-Term IGF-1 SDS Range in Relation to Cardiovascular Risk Factors and Vascular Parameters

Lower hs-CRP levels were observed in patients with upper-normal IGF-1 SDSs in comparison to those with lower-normal levels (Figure 1). Furthermore, hs-CRP levels showed a negative correlation with the 5-year mean IGF-1 SDS (r = −0.705, *p* < 0.001), as shown in Figure 1, and a positive correlation with body fat percentage (r = 0.383, *p* = 0.037), although the latter was only nominally significant. A trend of negative correlation between the mean IGF-1 SDS and body fat percentage was also observed over the 5-year period but did not reach statistical significance (r = −0.291, *p* = 0.141). In patients over 40 years of age, additional correlations of the 5-year mean IGF-1 SDS were observed with truncal fat mass (r = −0.474, *p* = 0.047) and HOMA-IR (r = −0.515, *p* = 0.029); however, their significance was lost after correction for multiple testing. Upper-normal IGF-1 SDSs were associated with lower body fat percentage (Figure 2); however, this association did not remain significant after adjusting for sex (*p* = 0.394). Lipid parameters and lean body mass showed no association with the 2-year IGF-1 SDS range or 5-year mean IGF-1 SDS. Markers of glucose metabolism, including HbA1c, HOMA IR, and glucose after OGTT, also had no association with any IGF-1 SDS parameters. Likewise, IGF-1 SDS parameters showed no association with IMT (5-year mean: r = 0.087, *p* = 0.666) or RHI (5-year mean: r = −0.032, *p* = 0.875).

### 2.4. Long-Term IGF-1 SDS Range in Relation to Adverse Effects of GHRT

A higher prevalence of adverse effects was reported in the group with lower-normal IGF-1 SDSs. In this group, 5 adverse events were reported by 4 patients (30.77%), including 3 reports of joint pain, 1 report of severe headache, and 1 report of ankle swelling. Among the patients with upper-normal IGF-1 SDSs, two (11.11%) experienced adverse events, both presenting with ankle swelling and one additionally reporting joint pain.

### 2.5. Long-Term IGF-1 SDS Range in Relation to Self-Reported Adherence

During the two months prior to inclusion, patient-reported adherence to GHRT differed slightly between the upper- and lower-normal IGF-1 SDS groups. Among patients with lower-normal IGF-1 SDSs, one patient (7.69%) reported at least three missed doses and one (7.69%) reported two missed doses, while the others (84.62%) missed one dose or fewer. In the upper-normal IGF-1 SDS group, 5 patients (27.78%) reported 3 or more omitted doses, 2 patients (11.11%) reported missing 2 doses, and the remaining 11 patients (61.11%) missed 1 dose or fewer over the period of two months.

## 3. Discussion

We demonstrated the association between the long-term maintenance of upper-normal IGF-1 SDSs and reduced markers of systemic inflammation, as well as the correlation between the 5-year mean IGF-1 SDS and hs-CRP. The critical role of GH in the modulation of inflammation is well established [14], as patients with AGHD consistently exhibit heightened inflammatory activity, marked by elevated levels of pro-inflammatory cytokines and hs-CRP compared to healthy individuals [15,16,17]. Decreased activity of hormone-sensitive lipase [18] and local cortisol excess [19] due to AGHD promote excessive lipid accumulation and visceral obesity [1]. These changes lead to an adverse adipokine profile and obesity-induced insulin resistance, both contributing to low-grade chronic inflammation [1,14]. On the other hand, GHD may exert direct effects on systemic inflammation through an imbalance between pro-inflammatory and anti-inflammatory gene expression profiles in macrophages [14]. For instance, Seki et al. demonstrated an inverse association between GH secretion and hs-CRP levels independently of other anterior pituitary hormones and metabolic complications of AGHD [20]. This is clinically reflected in findings that GHRT can significantly reduce inflammatory markers independently of changes in body composition [20,21,22]. In addition, a few studies, including ours, have established a negative correlation between IGF-1 and hs-CRP levels [4,17,21,23], supporting a direct effect of GHRT on low-grade inflammation while others did not detect this correlation [20,22,24]. However, a potential correlation of hs-CRP with body fat percentage was also identified, suggesting an interplay of these factors. This study highlights a combined effect of GHRT on systemic inflammation, mediated through both direct mechanisms and changes in body composition. Although the effect of GHRT on reducing hs-CRP has been recognized [21,22,23,24,25,26], this study is the first to demonstrate the association between long-term maintenance of IGF-1 SDS within the upper-normal range and anti-inflammatory effect of GHRT.

An association between sex and the parts of the normal IGF-1 SDS spectrum was observed in this study. Approximately three-quarters of men maintained IGF-1 SDSs within the upper-normal range over a 2-year period, whereas the same proportion of women achieved the lower-normal range. This observation is supported by studies on larger cohorts reporting that IGF-1 SDSs of female patients more frequently remained below the normal range [27,28,29,30]. The sex-based disparity in the achievement of parts of the normal range appears even more pronounced in our patient cohort. For comparison, Höybye et al. defined the upper-normal IGF-1 SDS as the target range and reported that one-third of female patients remained below this range [30], whereas in our study, three-quarters of the women were within the lower-normal range. The challenging titration of GHRT and a certain degree of GH resistance in women are attributed to estrogens’ attenuation of GH’s liver-mediated actions by inhibiting GH receptor signaling, thereby reducing IGF-1 production. While estrogens attenuate responsiveness to GH, androgens enhance it [31]. There are some observations that men derive greater benefits from GHRT than women, particularly in lipoprotein metabolism [30,32], body composition [33,34], and bone mineral density [27]. In addition, GHRT was shown to have a more favorable effect on mortality in male than female patients with hypopituitarism [35,36]. These findings may be partly attributed to a lower IGF-1 SDS in women in comparison to men with AGHD. Furthermore, we demonstrated that the reduction in systemic inflammation is greater in male patients, with a statistically significant 1.0 mg/L lower hs-CRP level compared to women. In general, higher hs-CRP levels were observed in untreated male patients [20], while no sex-related differences have yet been demonstrated in the anti-inflammatory effects of GHRT. Aside from the discussed difference in hs-CRP and physiological differences in HDL cholesterol and body fat percentage, no other significant discrepancies between sexes were identified in this study. However, our findings of a lower-normal IGF-1 SDS and elevated hs-CRP levels in women compared to men may support the hypothesis of the undertreatment of women with AGHD, influencing the GHRT outcome regarding anti-inflammatory effects.

Among patient characteristics, in addition to sex, the duration of GHRT was associated with the IGF-1 SDS range. The observed association between upper-normal IGF-1 SDSs and a prolonged duration of GHRT is consistent with the findings of Rochira et al., who grouped patients with AGHD by treatment duration and reported an increase in IGF-1 SDSs alongside decreasing GH doses in the group with the longest treatment duration [37]. This finding may be explained by the higher age of patients with a longer GHRT duration, as aging increases sensitivity to GH replacement [38]. Nevertheless, no significant association between age and IGF-1 SDS parameters was detected in this study. Over the 5-year period, the etiology of AGHD was also correlated with the mean IGF-1 SDS, which was significantly lower in patients with tumor-related compared to non-tumor-related etiology. Data on the differences in GH replacement in relation to the etiology of AGHD are scarce. Yuen et al. reported patients with craniopharyngioma required higher doses of GH yet still exhibited lower IGF-1 SDSs compared to patients with extrasellar tumors [39].

Despite the limited data comparing upper- and lower-normal IGF-1 SDS levels, there is some evidence regarding their effects on altering body composition. Van Bunderen et al. conducted two randomized studies to compare the effects of high-normal (SDS of 1 to 2) and low-normal (SDS of −2 to −1) IGF-1 ranges. The difference between the groups was significant for waist circumference in female patients, with an increase in IGF-1 levels leading to a decrease in waist circumference [40]. The waist circumference measurements in our study did not differ significantly between the groups; nevertheless, the group with upper-normal IGF-1 SDSs had a lower percentage of body fat compared to the lower-normal group. Since the upper-normal IGF-1 SDS group had a higher proportion of men, who physiologically exhibit a lower body fat percentage than women, a statistical adjustment for sex was performed. After this adjustment, the significance was lost, indicating that uneven sex distribution between the upper- and lower-normal range groups significantly contributes to the initially observed difference in fat percentage. This raises an important question: is the observed difference in fat percentage between the sexes solely attributable to physiological factors, or could it also reflect a more favorable body composition in male patients as an effect of GHRT and differences in IGF-1 levels? As previously suggested in the literature and discussed above, men may derive greater benefits from GHRT compared to women. However, our study could not address this issue, as the small number of participants limits further sex-specific subanalyses, and our cross-sectional study design does not allow for the determination of causality.

The study group of van Bunderen et al. published another study confirming the favorable effect of high-normal IGF-1 levels on waist circumference, while observing more pronounced induction of insulin resistance during short-term GHRT [12]. The present study identified no association of the long-term IGF-1 SDS range with the insulin resistance index, fasting glucose, or HbA1c, supporting the resolution of adverse effects on insulin resistance with prolonged GHRT [41,42]. Overall, fewer adverse events were reported in patients with a mean IGF-1 SDS within the upper-normal range. These findings confirm that targeting an upper-normal IGF-1 SDS range is safe as long as the upper limit is not exceeded, as emphasized in the guidelines [43]. Further research on the safety of dose titration to achieve the upper-normal range is certainly needed, as most adverse effects of GHRT are dose-dependent [10]. The risks of developing new cancers and new-onset diabetes should also be considered; nevertheless, the safety profile of GHRT appears favorable in long-term surveillance studies [43].

A meta-analysis of placebo-controlled trials on GHRT comparing low-dose and high-dose treatments highlighted a dose-related effect on increasing lean body mass and reducing fat mass. However, in contrast to effects on body composition, the impact of GHRT on the lipid profile showed no dose dependency, as both low- and high-dose treatments achieved comparable reductions in total and LDL cholesterol levels [13]. Similarly, in our study, no associations were established between lipids and IGF-1 SDS parameters, suggesting that upper IGF-1 levels may not provide additional improvements in lipid parameters. In this study, the IGF-1 SDS range predominantly exhibited associations or trends towards associations with body fat parameters, with no significant effects observed on lean body mass. Conversely, Zheng et al. reported a correlation between IGF-1 levels and the skeletal muscle index, a parameter of lean body mass; unlike the present study, however, their findings were based on GHRT with a fixed low GH dose [44]. This discrepancy in findings across different titration protocols suggests that GHRT-induced improvements in lean body mass may be achievable at lower doses. Meanwhile, GH doses titrated to maintain IGF-1 levels in the upper-normal range seem to contribute more significantly to reductions in body fat rather than further increases in lean body mass.

Although beneficial changes in cardiovascular profiles were associated with long-term IGF-1 SDS ranges, no associations were identified with markers of subclinical atherosclerosis. Neither the upper- vs. lower-normal IGF-1 SDS range nor the 5-year mean IGF-1 was directly associated with IMT or RHI.

The limitations of our study are inherent to the cross-sectional design, which does not provide conclusions on the causality of the observed associations. Moreover, potential associations between the IGF-1 SDS range and certain parameters, such as markers of subclinical atherosclerosis, would be more effectively identified in a prospective interventional study involving treatment-naïve patients. The number of subjects included is small, which limits the generalizability of our results. Furthermore, our findings regarding the benefits of upper-normal IGF-1 SDSs were discussed in the context of their effects on cardiovascular risks and diseases. van Bunderen et al. published some additional data on the differences in wellbeing when comparing high- and low-normal IGF-1 target levels. High-normal levels were associated with an improved quality of life in patients with AGHD [40] but were also related to impaired prefrontal cognitive function in female patients [45]. A major strength of our study is that we explored associations between the IGF-1 SDS range and the efficacy of GHRT over an extended period, analyzing the mean IGF-1 SDS measurements over 2- and 5-year periods.

## 4. Materials and Methods

Our cross-sectional study included 31 patients, managed at the Department of Endocrinology, Diabetes and Metabolic Diseases of University Medical Center Ljubljana. All patients were treated by daily GH derivatives for at least 5 years and had a 2-year mean IGF-1 SDS within the normal range, between −2 and +2. The diagnosis of AGHD was made in accordance with the Endocrine Society guidelines [10], GH Research Society guidelines [46], or other relevant guidelines applicable at the time of AGHD diagnosis. There were no restrictions on age, sex, disease onset time, or disease control status for inclusion. The exclusion criteria comprised active malignant diseases, evidence of growth of pituitary adenoma, or other benign intracranial tumors within the last year before inclusion, acute illness, weight loss of more than 5% within the past 180 days, chronic liver or kidney disease, diabetes mellitus, active Cushing’s syndrome within the past 24 months, and history of acromegaly. The study protocol was approved by the Slovenian National Medical Ethics Committee (approval number 0120-138/2022/6) and conducted according to the guidelines of the Declaration of Helsinki and its later amendment. Informed written consent to participate in this study and to have their data published in a journal article was obtained from all participants.

### 4.1. Clinical Characteristics

Data on the etiology of pituitary disease and its time of onset (child- or adult-onset AGHD), GHRT duration, replacement therapy for other pituitary hormones, arterial hypertension, and treatment with antilipemic agents were derived from medical documentation. The patients’ medical history regarding smoking habits was obtained. Adherence to GHRT was evaluated through the patient-reported numbers of missed doses during the two-month period prior to study inclusion.

### 4.2. Anthropometric Measurements and Body Composition Assessment

Height was measured by a wall-mounted stadiometer with an accuracy of 1 cm; weight was measured by a standard body scale with an accuracy of 0.5 kg. Waist circumference was measured at the midpoint between the lower margin of the last palpable ribs and the top of the iliac crest. Hip circumference was measured around the widest portion of the buttocks, with the tape parallel to the floor. Waist-to-hip ratio was calculated from waist and hip circumference measurements. Body mass index (BMI) was calculated as weight in kilograms divided by height in meters squared. Body composition was assessed via dual-energy X-ray absorptiometry (DXA) (Discovery DXA System (Hologic, Bedford, MA, USA)).

### 4.3. Biochemical and Hormone Analysis

Venous blood samples were obtained from subjects in the morning, following an overnight fast for the measurement of total cholesterol (TC), low-density lipoprotein cholesterol (LDL), high-density lipoprotein cholesterol (HDL), triglyceride (TG), IGF-1, high-sensitivity C-reactive protein (hs-CRP), and glycosylated hemoglobin (HbA1c). Serum glucose and insulin levels were measured at 0 and 120 min post 75 g oral glucose tolerance test (OGTT). TC and TG were measured with the enzyme method, LDL and HDL with the method of elimination/chatalasis (ADVIA^®^ Chemistry systems, Siemens Healthcare, Erlangen, Germany). Glucose was measured with the standard oxidase method (Beckman Coulter Glucose Analyzer, Beckman Coulter Inc., Brea, CA, USA). Serum insulin was measured with the two-site sandwich, chemiluminiscent immunoassay and the “Atellica IM Insulin (IRI)” kit and (Atellica IM 1600 analyzer, Minaris Medical Co. for Siemens Healthcare Diagnostics, Mountain View, CA, USA). HbA1c was measured using capillary electrophoresis (Capillarys 2 FLEX-PIERCING, Sebia, Lisses, France). IGF-1 was measured with the chemilumiscent method (CLIA) on an iSYS analyzer (IDS-iSYS Insulin-like Growth Factor-I Immunodiagnostic Systems Limited, Boldon, UK). hs-CRP was measured using a high-sensitivity immunoturbidimetric assay on a Cobas^®^ 8000 modular analyzer (Roche Diagnostics, Mannheim, Germany). Insulin resistance was assessed by the Homeostatic Model Assessment for Insulin Resistance (HOMA-IR), calculated as HOMA-IR = [insulin (mU/L) × glucose (mmol/L)]/22.5.

The mean IGF-1 SDS was calculated using measurements taken during 2-year and 5-year periods prior to study enrolment. IGF-1 SDS measurements were performed during routine endocrinological examinations at our center as part of GHRT monitoring. Typically, one measurement was taken annually, with an additional measurement conducted at the time of study inclusion. In a few patients with more frequent endocrinological visits, more than one measurement per year was available. Consequently, for most participants, the 2-year mean IGF-1 SDS was calculated from three measurements, while the 5-year mean was derived from six measurements. Patients were grouped according to their 2-year mean IGF-1 SDS into an upper-normal range, defined as 0 to +2, or a lower-normal IGF-1 SDS range, defined as −2 to 0. The 5-year mean IGF-1 SDS was analyzed as a continuous variable.

### 4.4. Vessel Morphology and Function Assessment

IMT was assessed using carotid ultrasound examination following a protocol previously described elsewhere [7]. The mean IMT of each participant was calculated as an average of four measurements, including the right and left side of two segments: at the common carotid arteries 1 cm below bifurcation and at the carotid bifurcation. Endothelial function was assessed using peripheral vasodilator response measurements from a fingertip pulse amplitude tonometry device. An Endopat 2000 device (Itamar Medical Ltd., Caesarea, Israel) was used for plethysmographic recordings of finger arterial pulse wave amplitude. The participants were in a semi-seated position with extended forearms at a 40° angle. Pneumatic probes were placed on both index fingers and were inflated to 70 mmHg. After measuring baseline conditions (5 min), a cuff was placed on the left forearm and was inflated to 60 mmHg above the systolic pressure (5 min) and then rapidly deflated to induce the reactive hyperemia phase (recorded for 5 min). The reactive hyperemia index (RHI) was automatically calculated using the ratios between pulse wave amplitudes during the reactive hyperemia and baseline phase.

### 4.5. Statistical Analysis

To describe the distribution of continuous variables, we used the median with the interquartile range, while categorical variables were described using frequencies. The Shapiro–Wilk test was used to assess the normality of the distributions. As not all variables were normally distributed, non-parametric tests were employed for statistical analysis. Spearman’s rank correlation coefficient was used to analyze correlations between the continuous variables. The Mann–Whitney U test was used to compare the distribution of the continuous variables between different groups. Fisher’s exact test was used for comparing the distribution of the categorical variables. The main and interaction effects of the categorical variables on a continuous dependent variable was determined using an ANCOVA test. To reduce the likelihood of false positives, we applied the Bonferroni correction: with 25 parameters compared, we set the threshold for statistical significance at ≤0.002 (0.05/25), while *p*-values between 0.002 and 0.05 were considered nominally significant. All statistical analyses were performed using IBM SPSS Statistics, version 27.0 (IBM Corporation, Armonk, NY, USA).

## 5. Conclusions

In conclusion, our findings suggest an association between the long-term maintenance of IGF-1 SDS in the upper-normal IGF-1 SDS range and lower markers of systemic inflammation, a characteristic feature of AGHD. Additionally, male patients were more likely to achieve the upper-normal range compared to female patients, highlighting the importance of careful GHRT dose titration in women. Prospective randomized studies are needed to evaluate the long-term and sex-specific effects and safety of targeting the upper- vs. lower-normal IGF-1 SDS range in AGHD, for both daily and long-acting GH derivatives. This topic is particularly important to address, as findings regarding the optimal IGF-1 SDS range for GH replacement would have great potential for clinical application.

## Figures and Tables

**Figure 1 ijms-26-02010-f001:**
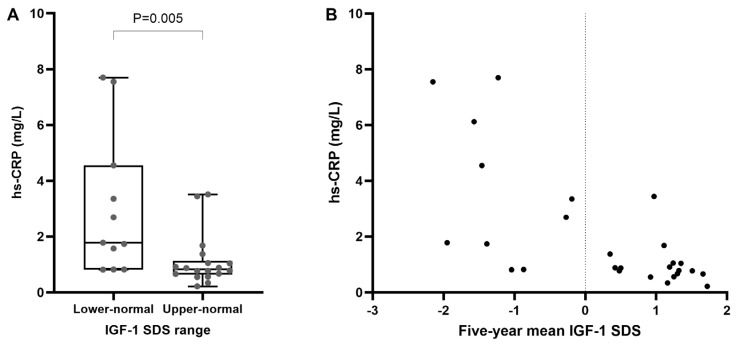
Association between IGF-1 SDS and inflammation marker (hs-CRP). (**A**): Distribution of hs-CRP levels in patients categorized into lower-normal (−2 to 0) and upper-normal (0 to +2) IGF-1 SDS ranges. (**B**): Negative correlation between 5-year mean IGF-1 SDS and hs-CRP levels. Legend: hs-CRP—high-sensitive C-reactive protein, IGF-1—insulin-like growth factor 1, SDS—standard deviation score.

**Figure 2 ijms-26-02010-f002:**
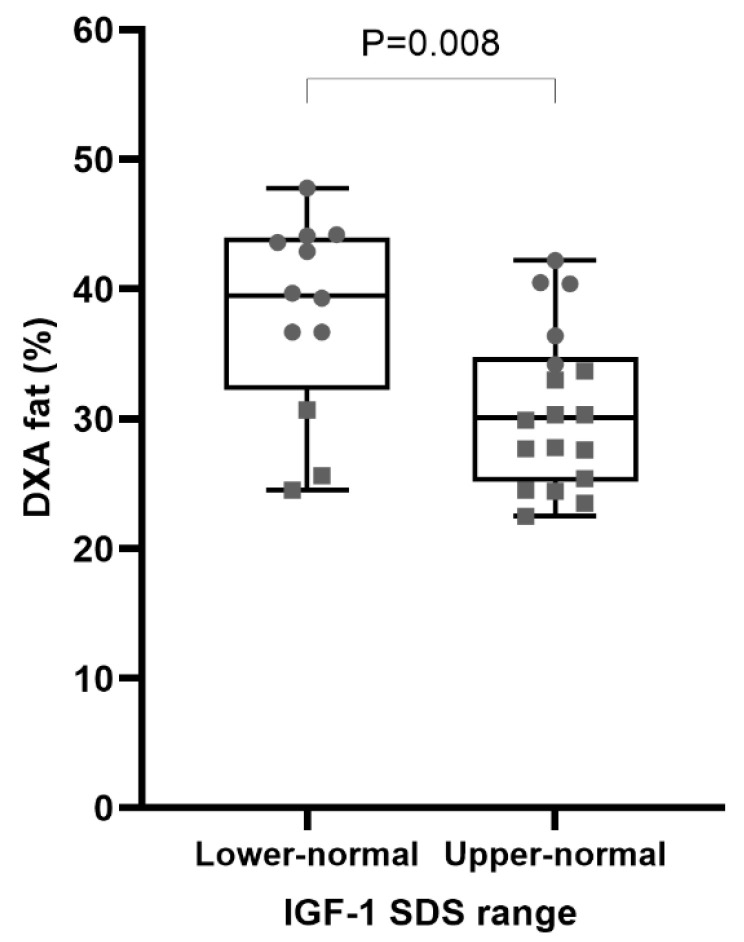
Association between IGF-1 SDS range and body fat percentage. The association between IGF-1 SDS range and body fat percentage was statistically significant (*p* = 0.008); however, after adjusting for sex, the significance was lost (*p* = 0.394), indicating that sex played a key role in this relationship. To visually depict the influence of sex, different symbols are used: squares represent male values, while circles represent female values. Legend: DXA—dual-energy X-ray absorptiometry, IGF-1—insulin-like growth factor 1, SDS—standard deviation score.

**Table 1 ijms-26-02010-t001:** Baseline characteristics.

Parameter	Category/Unit	Values
Sex	Male, N (%)	17 (45.2)
	Female, N (%)	14 (54.8)
Age	Years, mean (range)	46.3 (22–68)
Etiology of AGHD	Tumor-related, N (%)	23 (74.2)
	Non-tumor, N (%)	8 (25.8)
Onset of AGHD	Child-onset, N (%)	18 (58.1)
	Adult-onset, N (%)	13 (41.9)
BMI [2]	kg/m^2^, median (25–75%)	27.9 (23.3–31)
WHR [2]	cm/cm, median (25–75%)	0.91 (0.82–1)
Fat percentage	%, median (25–75%)	33 (25.6–40.4)
Cholesterol	mmol/L, median (25–75%)	5.0 (4.5–5.9)
HDL cholesterol	mmol/L, median (25–75%)	1.2 (1.1–1.5)
LDL cholesterol	mmol/L, median (25–75%)	3.6 (2.7–4.6)
Triglycerides	mmol/L, median (25–75%)	1.5 (1.0–2.0)
hs-CRP	mg/L, median (25–75%)	1.0 (0.8–2.9)
HbA1c	%, median (25–75%)	5.3 (5.2–5.5)
Glucose 0 min	mmol/L, median (25–75%)	5.0 (4.5–5.4)
Glucose 120 min	mmol/L, median (25–75%)	6.1 (5.2–8.0)
Insulin 0 min	μU/mL, median (25–75%)	8.7 (6.0–9.4)
Insulin 120 min	μU/mL, median (25–75%)	37.3 (27.3–65.9)
HOMA IR	Median (25–75%)	1.8 (1.3–2.2)
Active or former smoker [2]	No, N (%)	23 (79.3)
	Yes, N (%)	6 (20.7)
Arterial hypertension	No, N (%)	26 (83.9)
	Yes, N (%)	5 (16.1)
Statin therapy [1]	No, N (%)	22 (73.3)
	Yes, N (%)	8 (26.7)
Duration of GHRT	Years, median (25–75%)	20 (8–33.5)
IMT	mm, median (25–75%)	0.62 (0.56–0.77)
RHI	Median (25–75%)	2.02 (1.52–2.37)
2-year IGF-1 SDS range	No, N (%)	13 (41.9)
	Yes, N (%)	18 (58.1)
5-year mean IGF-1 SDS	Median (25–75%)	0.50 (−1.04–1.25)

Legend: N—normal, AGHD—adult growth hormone deficiency, BMI—body mass index, WHR—waist-to-hip ratio, HDL—high-density lipoprotein, LDL—low-density lipoprotein, hs-CRP—high-sensitivity C-reactive protein, HbA1c—glycated hemoglobin, min—minutes, HOMA IR—homeostatic model assessment of insulin resistance, GHRT—growth hormone replacement therapy, IMT—intima-media thickness, RHI—reactive hyperemia index, IGF-1—insulin-like growth factor 1, SDS—standard deviation score, [ ]—number of missing data points.

**Table 2 ijms-26-02010-t002:** Associations of IGF-1 SDS range with patient characteristics, cardiovascular risk factors, and vascular parameters.

Parameter	Category/Unit	Lower-Normal	Upper-Normal	*p*-Value
Sex	Male, N (%)	3 (25)	13 (72.2)	0.024 *
	Female, N (%)	9 (75)	5 (27.8)	
Age	Years, median (25–75%)	42 (27.3–56.8)	50.5 (39.5–58.5)	0.267
Etiology	Tumor-related, N (%)	11 (91.7)	11 (61.1)	0.099
	Non-tumor, N (%)	1 (8.3)	7 (38.9)	
Onset of AGHD	Child-onset, N (%)	6 (50)	12 (66.7)	0.458
	Adult-onset, N (%)	6 (50)	6 (33.3)	
BMI	kg/m^2^, median (25–75%)	28.5 (21.7–31.6)	27.9 (23.6–30.4)	1.000
WHR	cm/cm, median (25–75%)	0.86 (0.81–0.93)	0.96 (0.85–1.02)	0.191
Fat percentage	%, median (25–75%)	39.5 (32.2–44.0)	30.1 (25.2–34.8)	0.008 *
Cholesterol	mmol/L, median (25–75%)	5.0 (4.5–5.9)	5.1 (4.6–6.1)	0.723
HDL cholesterol	mmol/L, median (25–75%)	1.2 (1.1–1.5)	1.3 (1.1–1.4)	0.851
LDL cholesterol	mmol/L, median (25–75%)	3.8 (2.9–4.5)	3.5 (2.7–4.7)	0.983
Triglycerides	mmol/L, median (25–75%)	1.6 (1.1–2.0)	1.5 (0.8–2.5)	0.950
hs-CRP	mg/L, median (25–75%)	1.8 (0.8–4.6)	0.8 (0.6–1.1)	0.005 *
HbA1c	%, median (25–75%)	5.3 (5.2–5.5)	5.3 (5.1–5.6)	0.755
Glucose 120 min	mmol/L, median (25–75%)	6.2 (5.4–7.7)	5.9 (4.6–7.8)	0.545
HOMA IR	Median (25–75%)	2.0 (1.7–2.3)	1.6 (1.2–2.2)	0.285
Active or former smoker	No, N (%)	9 (81.8)	13 (76.5)	1.000
	Yes, N (%)	2 (18.2)	4 (23.5)	
Arterial hypertension	No, N (%)	11 (91.7)	15 (83.3)	0.632
	Yes, N (%)	1 (8.3)	3 (16.7)	
Statin therapy	No, N (%)	10 (83.3)	11 (64.7)	0.408
	Yes, N (%)	2 (16.7)	6 (35.3)	
IMT	mm, median (25–75%)	0.63 (0.47–0.78)	0.63 (0.57–0.76)	0.692
RHI	Median (25–75%)	2.09 (1.76–2.43)	1.94 (1.45–2.31)	0.415
Duration of GHRT	Years, median (25–75%)	15 (4–20)	28 (16.5–35)	0.033 *

Legend: N—normal, AGHD—adult growth hormone deficiency, BMI—body mass index, WHR—waist-to-hip ratio, HDL—high-density lipoprotein, LDL—low-density lipoprotein, hs-CRP—high-sensitivity C-reactive protein, HbA1c—glycated hemoglobin, min—minutes, HOMA IR—homeostatic model assessment of insulin resistance, IMT—intima-media thickness, RHI—reactive hyperemia index, GHRT—growth hormone replacement therapy, *—statistically significant *p*-value.

## Data Availability

The data collected in this study are available for review upon reasonable request.

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
