# Peer review of "Long-Term IGF-1 Maintenance in the Upper-Normal Range Has Beneficial Effect on Low-Grade Inflammation Marker in Adults with Growth Hormone Deficiency"

_ijms, 2025, doi:10.3390/ijms26052010_

Round 1
Reviewer 1 Report
Comments and Suggestions for Authors
Introduction:
The hypothesis should be clearly articulated to outline the study’s primary objectives.
Methods:
The study includes only 31 patients, which is a relatively small sample. The authors should justify whether this sample size is statistically adequate for drawing conclusions.
The exclusion criteria are well-described; however, the method of participant selection is unclear. Were any patients lost to follow-up, and if so, how might this have impacted the results?
Additionally, please specify how frequently IGF-1 was measured throughout the study period and how consistent these measurements were across patients.
Did female patients use oral estrogens, and if so, did that influence their IGF levels?
Was adherence to GHRT considered in this study? If not, this should be acknowledged as a limitation.
Results:
While sex differences are noted in IGF-1 SDS maintenance, further explanation of potential hormonal interactions affecting GH response is needed. Were female patients on estrogen therapy, and if so, could this have influenced their lower IGF-1 SDS levels?
The study suggests no association between IGF-1 SDS and markers of subclinical atherosclerosis (IMT, RHI). However, previous literature indicates that GH influences vascular function. Could the small sample size have limited the detection of significant differences?
The study notes that the upper-normal IGF-1 SDS group had lower body fat percentages. However, were there any changes in muscle mass or metabolic rate that could further explain this improvement?
Discussion:
The study could benefit from a more in-depth comparison with research examining different GH dosing strategies. I recommend providing specific recommendations for clinicians regarding GHRT dose adjustments to maintain IGF-1 in the upper range.
Author Response
We sincerely thank Reviewer 1 for helpful comments and suggestions, which we incorporated into the revised version as follows:
Comment 1: Introduction: The hypothesis should be clearly articulated to outline the study’s primary objectives.
Response 1: Thank you for the suggestion. We incorporated our hypothesis in the final paragraph of the introduction. The sentence reads as follows (please see page 2, lines 75-77; numbering applies to the marked version of the Manuscript with the 'Track Changes' function enabled):
“We hypothesized that patients with upper-normal IGF-1 SDS would exhibit a more favorable cardiovascular profile compared to those with IGF-1 SDS within low-er-normal range.”
Comment 2: The study includes only 31 patients, which is a relatively small sample. The authors should justify whether this sample size is statistically adequate for drawing conclusions.
Response 2: Thank you for your notion. Due to the rarity of growth hormone deficiency in adulthood, the pool of patients available for study inclusion is generally limited. With the sample of 31 patients in this study, we included nearly half of the patients with AGHD who are currently treated by GHRT. Furthermore, the number of participants in this study, as well as in similar studies, is somewhat limited by the applied methodology, particularly ultrasound assessment of intima-media thickness (IMT) and the evaluation of endothelial function using the EndoPAT 2000 system. The sample size in our study is consistent with other similarly designed studies in this field. For example, here are a few key publications in the field of cardiovascular health in patients with AGHD: McCallum et al. included 16 patients in the study on hs-CRP (doi: 10.1111/j.1365-2265.2005.02245.x), Smith et al. demonstrated effect of GHRT on endothelial function on 32 patients (doi: 10.1046/j.1365-2265.2002.01514.x), Pfeifer et al. included 11 patients in the study demonstrating the effect of GHRT on reversing early atherosclerotic changes (doi: 10.1210/jcem.84.2.5456). Even the largest centers dedicated to IGF-1 research, which have made major contributions to the field, employ sample sizes in their studies that are comparable to ours. For instance, the study group of Colao et al. included 34 patients in the cross-over study regarding IMT (doi: 10.1210/jc.2004-2247), 36 patients in the study on gender-related effects of GHRT (doi: 10.1210/jc.2005-0597), and 23 patients in the prospective study on withdrawal and restarting of GHRT (doi: 10.1210/jc.2004-1844). Based on this, we consider our sample size sufficient to draw relevant conclusions in the field of GHRT treatment for patients with AGHD.
Comment 3: The exclusion criteria are well-described; however, the method of participant selection is unclear. Were any patients lost to follow-up, and if so, how might this have impacted the results?
Response 3: Thank you for your question. Participants were selected according to the following selection protocol: among patients who had been managed at our centre and were receiving GHRT for at least 5 years, we selected those willing to participate in the study. If their 2-year mean IGF-1 standard deviation score (SDS) fell within normal range (-2 to +2 SD) and the remaining inclusion criteria were met, they were considered suitable for inclusion in this study. As the study was cross-sectional in nature, no participants were lost during the follow-up.
Comment 4: Additionally, please specify how frequently IGF-1 was measured throughout the study period and how consistent these measurements were across patients.
Response 4: Thank you for the suggestion. IGF-1 SDS measurements were conducted during regular endocrinological examinations at our centre as part of the GHRT monitoring. Commonly one measurement was performed annually, with an additional measurement taken at the time of study inclusion. For few patients who had more frequent endocrinological visits, more than one measurement was taken per year. Therefore, for the majority of participants, the 2-year mean was calculated from three measurements, while the 5-year mean was calculated from six measurements of IGF-1 SDS. We have added information on this topic in the Materials and Methods section (please see page 13, lines 397-403):
“IGF-1 SDS measurements were performed during routine endocrinological examinations at our center as part of GHRT monitoring. Typically, one measurement was taken annually, with an additional measurement conducted at the time of study inclusion. In a few patients with more frequent endocrinological visits, more than one measurement per year was available. Consequently, for most participants, the 2-year mean IGF-1 SDS was calculated from three measurements, while the 5-year mean was derived from six measurements.”
Comment 5: Did female patients use oral estrogens, and if so, did that influence their IGF levels?
Response 5: Thank you for your question. Two thirds of women in our cohort used oral estrogens, namely 9 out of 14 patients (64.29 %). Patients using oral estrogens were evenly distributed between the two groups according to IGF-1 SDS range: 4 in the upper-normal and 5 in the lower-normal range. However, in the group with lower-normal IGF-1 SDS, which is smaller, women receiving oral estrogens represented a higher proportion (38.46 %) compared to their proportion in the upper-normal range group (22.22 %). On the other hand, women with no estrogen replacement were less evenly distributed, with 4 out of 5 in the lower-normal IGF-1 SDS range; however, it is possible that this distribution is merely coincidental. Unfortunately, we believe that this group of 5 women (and, in general, the group of 14 women in this study) is too small for further statistical subanalyses regarding the association between oral estrogens and their IGF-1 levels.
We appreciate your emphasis on the issue of oral estrogen use in female patients receiving GHRT. We believe that this information is highly relevant and will add value to the article, which is why we have included it in the Results section under the subheading “2.1 Patient characteristics”. The added sentence reads as follows (please see page 3, lines 91-93):
“Among the 14 female patients, all 9 premenopausal women were receiving oral estrogen replacement, while none of the postmenopausal women were on hormone replacement therapy.”
Additional information on the distribution of women between upper- and lower-normal range groups according to oral estrogen replacement therapy is provided in subsection “2.2. Long-term IGF-1 SDS range in relation to patient characteristics” (please see page 5, lines 122-126):
“Patients receiving oral estrogens were relatively evenly distributed between the two IGF-1 SDS range groups, with 4 in the upper-normal and 5 in the lower-normal range. On the other hand, women not receiving estrogen replacement were less evenly distributed, with 4 out of 5 in the lower-normal IGF-1 SDS range.”
Comment 6: Was adherence to GHRT considered in this study? If not, this should be acknowledged as a limitation.
Response 6: Thank you for your question and suggestion. We indeed collected data on adherence by assessing patient-reported missed doses. At the time of study inclusion patients were asked how many doses they had missed in the past two months, with response options of 3 or more, two, or one or fewer doses. We have added information on adherence in the Results where we introduced a new subsection entitled “2.5. Long-term IGF-1 SDS range in relation to the adherence”, which reads as follows (please see pages 8-9, lines 184-191):
“2.5. Long-term IGF-1 SDS range in relation to self-reported adherence
During the two months prior to inclusion, patient-reported adherence to GHRT differed slightly between the upper- and lower-normal IGF-1 SDS groups. Among patients with lower-normal IGF-1 SDS, one patient (7.69 %) reported at least 3 missed doses and one (7.69 %) reported two missed doses, while the others (84.62 %) missed one dose or fewer. In the upper-normal IGF-1 SDS group, 5 patients (27.78 %) reported 3 or more omitted doses, two patients (11.11 %) reported missing two doses, and the remaining 11 patients (61.11 %) missed one dose or fewer over the period of two months.”
Information on the adherence assessment method was added to the Materials and Methods under the subsection 4.1 Clinical characteristics. The sentence reads as follows (please see page 12, lines 364-366):
“Adherence to GHRT was evaluated through the patient-reported number of missed doses during the two-month period prior to study inclusion.”
Comment 7: While sex differences are noted in IGF-1 SDS maintenance, further explanation of potential hormonal interactions affecting GH response is needed. Were female patients on estrogen therapy, and if so, could this have influenced their lower IGF-1 SDS levels?
Response 7: Thank you for the suggestion. We provided some additional information on possible mechanisms explaining growth hormone resistance in female patients with AGHD. The question on estrogen therapy is answered in the reply under the question number 5. Some data on estrogens’ interaction with growth hormone was added to the Discussion section, while the previous sentence was adjusted and now reads as follows (please see page 9, lines 228-231):
“The challenging titration of GHRT and a certain degree of GH resistance in women are attributed to estrogens’ attenuation of GH’s liver-mediated actions by inhibiting GH receptor signalling, thereby reducing IGF-1 production. While estorgens attenuate responsiveness to GH, androgens enhance it [32].”
Comment 8: The study suggests no association between IGF-1 SDS and markers of subclinical atherosclerosis (IMT, RHI). However, previous literature indicates that GH influences vascular function. Could the small sample size have limited the detection of significant differences?
Response 8: Thank you for the question. The sample size in this study may partially contribute to the lack of significant associations between IGF-1 SDS parameters and markers of subclinical atherosclerosis. In our view, the primary reason lies in the fact that we assessed vascular structure and endothelial function in patients who have been receiving well-titrated GHRT for an extended period. Studies that determined the impact of GHRT on vascular structure and endothelial function so to mostly observed in treatment-naïve patients in a period of only few months after GHRT initiation (doi: ). The impact of GHRT on vascular parameters would undoubtedly be more apparent in an interventional study, where the effects of initiating GHRT would be studied in treatment-naïve patients with AGHD. Therefore, we believe that the limitations in detecting a potentially significant associations with vascular parameters arise from both the sample size, as you highlighted, as well as the cross-sectional design of the study and the inclusion of non-treatment-naïve participants undergoing long-term GHRT. We highlighted this in the following sentence, which has been added to the Discussion section in the paragraph on this study’s limitations (please see page 11, lines 323-326):
“Moreover, potential associations between the IGF-1 SDS range and certain parameters, such as markers of subclinical atherosclerosis, would be more effectively identified in a prospective interventional study involving treatment-naïve patients.”
Comment 9: The study notes that the upper-normal IGF-1 SDS group had lower body fat percentages. However, were there any changes in muscle mass or metabolic rate that could further explain this improvement?
Response 9: Thank you for the question. Muscle mass was estimated indirectly as the largest compartment of the lean body mass, which was assessed by DXA. Lean body mass compartment includes all non-fat, non-bone tissues (primarily muscle, organs, water, and connective tissue). Unfortunately, we did not perform measurements that would assess basal metabolic rate or total daily energy expenditure, but this would definitely be an interesting idea for future research.
Regarding the lower body fat percentage in the group with upper-normal IGF-1 SDS, we performed an additional statistical analysis at the initiative of Reviewer 2, due to the significantly higher proportion of male patients in this group. By performing an ANCOVA statistical test, we verified whether the association between IGF-1 range and body fat percentage would still be significant after adjusting for sex, and it turned out that significance was lost in this subanalysis. This shows that the uneven sex distribution between the upper- and lower-normal range groups significantly contributes to the initially observed difference in fat percentage. We have tried carefully to address, explain, and comment this additional results in the Discussion. However, after performing this additional analysis, we have also changed the Title, Abstract, Results, Statistical analysis in Materials and Methods, and Figures accordingly.
Comment 10: The study could benefit from a more in-depth comparison with research examining different GH dosing strategies. I recommend providing specific recommendations for clinicians regarding GHRT dose adjustments to maintain IGF-1 in the upper range.
Response 10: Thank you for the suggestion. In our clinical practice, the treatment of AGHD follows an individualized titration approach, aiming to maintain IGF-1 SDS within the normal range. Other GHRT dosing strategies, such as fixed low doses or weight- and body surface area-based dosing, are no longer standard in clinical practice across Europe and the USA. Weight-based dosing, in particular, has been associated with supraphysiological GH doses, leading to an increased risk of adverse effects in adult patients with GHD. In contrast, individualized stepwise titration has demonstrated superior treatment efficacy, better adherence, and a lower incidence of side effects. Furthermore, current international guidelines (doi: 10.4158/GL-2019-0405) recommend individualized GHRT dose titration based on IGF-1 levels, clinical response, and side effect monitoring as the preferred dosing strategy for AGHD. Given that our study focuses on IGF-1 SDS as a primary objective, we believe it is most appropriate to compare our findings with studies using the same evidence-based titration approach. Additionally, alternative dosing strategies place significantly less emphasis on IGF-1 SDS as a treatment target, making direct comparisons less relevant to the scope of our investigation.
Due to the sample size of 31 patients, we do not feel confident in providing specific recommendations for clinicians regarding GHRT dose adjustments. However, we would like to emphasize this issue to encourage further discussion, consideration, and research in this field. If future studies confirmed our findings or reported similar results, a recommendation could be developed suggesting that targeting the upper-normal range of IGF-1 SDS in GHRT dosing may yield benefits. Most importantly, there is a need for studies that focus in-depth on the long-term safety of such dosing before recommending general targeting of the upper-normal range.
We appreciate the opportunity to clarify this aspect of our study and hope this explanation provides additional context.

Reviewer 2 Report
Comments and Suggestions for Authors
The authors conducted a cross-sectional study with 31 patients with AGHD receiving growth hormone replacement therapy (GHRT) with daily GH > 5 years and 2-year mean IGF-1 SDS between -2 to +2 SDS range. Patients were categorized according to their 2-year mean IGF-1 SDS into the upper- or lower-normal range. Associations of clinical characteristics, anthropometric parameters, laboratory tests, and vascular markers of subclinical atherosclerosis with the 2-year IGF-1 SDS range and 5-year mean IGF-1 SDS were explored. They found that long-term maintenance of upper-normal IGF-1 SDS was more frequent in men and in patients with longer GHRT. Patients with tumor-related AGHD had a lower 5-year mean IGF-1 SDS. Long-term maintenance of IGF-1 SDS in the upper-normal range was associated with lower high-sensitivity C-reactive protein (hs-CRP) levels and lower body fat. A negative correlation was identified between a hs-CRP and the 5-year mean IGF-1 SDS. They concluded that long-term maintenance of upper-normal IGF-1 SDS was associated with reduced low-grade inflammation and lower body fat.
The topic is highly important, but there are some problems with the interpretation of the data.
Comments:
- As the authors mentioned in the Discussion, men physiologically exhibit a lower body fat percentage than women. Therefore, the cause of the lower body fat percentage in the Upper-normal group can be the higher ratio of males in this group compared to the lower-normal group. Therefore, further statistical analysis would be necessary to clarify the role of gender. Otherwise, this conclusion cannot be accepted, consequently, the title, the abstract and the manuscript must be modified.
- Regarding the hs-CRP values, the ratio of tumor-related cases is higher in the lower-normal group compared to the upper-normal group. Were these patients apparently tumor-free? If not, the tumor mass may also contribute to the higher hs-CRP levels.
- Female and male patients could be marked on Fig 1 (and Fig 2) with different symbols (i.e.. ● vs ☐)
- Conclusions should be summarized in a separate section after Materials and Methods.
Author Response
We sincerely thank Reviewer 2 for helpful comments and suggestions, which we incorporated into the revised version as follows:
Comment 1: As the authors mentioned in the Discussion, men physiologically exhibit a lower body fat percentage than women. Therefore, the cause of the lower body fat percentage in the Upper-normal group can be the higher ratio of males in this group compared to the lower-normal group. Therefore, further statistical analysis would be necessary to clarify the role of gender. Otherwise, this conclusion cannot be accepted, consequently, the title, the abstract and the manuscript must be modified.
Response 1: We appreciate you highlighting the potential role of sex in body fat proportion, as there is uneven sex distribution between the upper- and lower-normal range groups. ANCOVA statistical test was performed to verify whether the association between IGF-1 range and body fat percentage would still be significant after adjusting for sex. Following this statistical analysis the association did not remain significant, with the p-value of 0.394. This shows that the uneven sex distribution between the upper- and lower-normal range groups contributes significantly to the initially observed difference in fat percentage. This raises a new question: whether the difference in fat percentage between sexes is purely due to physiological factors or if it could partially reflect more favorable body composition changes in male compared to female patients as a result of GHRT and differences in IGF-1 levels. As already proposed in the literature, men may derive greater benefits from GHRT, including its effects on altering body composition (doi: 10.1016/s0026-0495(99)90077-x, doi: 10.1093/ejendo/lvae060). However, due to the cross-sectional design of our study and the sample size, we are unable to address this question within the current research. Nonetheless, we have included it in the Discussion to encourage further investigation into sex-related differences in GHRT effects, particularly in the context of optimizing GHRT dosing. Based on these new insights, we have adjusted the title and abstract to reduce the emphasis on body fat and revised the sections on Statistical Analysis, Results, and Figures accordingly. Please refer to the revised Manuscript for these updates, all changes in the text are traceable. We appreciate the opportunity to clarify this aspect of our study
Comment 2: Regarding the hs-CRP values, the ratio of tumor-related cases is higher in the lower-normal group compared to the upper-normal group. Were these patients apparently tumor-free? If not, the tumor mass may also contribute to the higher hs-CRP levels.
Response 2: Thank you for your question. We agree that tumor mass would highly likely contribute to low-grade systemic inflammation. We reviewed medical records for all patients classified with tumor etiology to determine whether they had undergone surgery and if any residual or relapse of the tumor is present postoperatively. The absence of tumor growth had to be documented by two post-surgery magnetic resonance imaging, with the most recent scan performed ≤24 months prior to study inclusion. We established that all participants with tumor-related etiology had undergone surgical treatment and remained free of residual tumor or disease recurrence postoperatively. Due to the potential impact of tumor mass on hs-CRP, as you highlighted, we found it relevant to include the following information in the Results section. Previous sentence regarding GHD etiology was therefore slightly altered and now reads as follows (please see page 2, lines 86-89; numbering applies to the marked version of the Manuscript with the 'Track Changes' function enabled):
“Etiologically, 8 patients had non-tumor-related GHD, while 23 patients had tumor-related etiology., among which 4 patients had craniopharyngioma. None of the latter had residual or recurrent tumors, and four had a history of craniopharyngioma.”
Comment 3: Female and male patients could be marked on Fig 1 (and Fig 2) with different symbols (i.e. ● vs ☐).
Response 3: Thank you for the suggestion. We have added separate symbols for both sexes in the Figure 2 (previously named Figure 1), presenting fat percentage on DXA in relation to IGF-1 range. We found it very suitable for the graphical presentation of the parameter fat percentage, as it was initially significantly associated with IGF-1 range, but the significance was lost after adjusting for sex. We also mentioned this in the Figure 2 description to explain the necessary use of gender-specific symbols (please see page 8, line 170-175).
Comment 4: Conclusions should be summarized in a separate section after Materials and Methods.
Response 4: Thank you for the suggestion. The paragraph on conclusions has been placed after the section of Materials and Methods. It has also been slightly modified to put less emphasis on body composition and more emphasis on the role of sex.
To inform you, we have included additional information on the adherence, obtaining of IGF-1 measurements, oral estrogen therapy in female patients, and an explanation of the interaction between estrogens and growth hormone at the initiative of Reviewer 1.

Round 2
Reviewer 1 Report
Comments and Suggestions for Authors
The authors sufficiently addressed all questions, and the article could be published in its current form.
Reviewer 2 Report
Comments and Suggestions for Authors
The manuscript was significantly improved.